# FASD: The Living Experience of People with Fetal Alcohol Spectrum Disorder—Results of an Anonymous Survey

**Emily Hargrove [1], C. J. Lutke [2], Katrina Griffin [2], Myles Himmelreich [2], Justin Mitchell [2], Anique Lutke [2] and Peter Choate [3,*]**

[1] Faculty of Graduate Studies, Walden University, Minneapolis, MN 55401, USA; emily.hargrove@waldenu.edu
[2] The Adult Leadership Committee on FASD, Vancouver, BC, Canada; c.lutke2015@gmail.com (C.J.L.); 2323@telus.net (M.H.); jlutke@shaw.ca (J.M.); alutke24@gmail.com (A.L.)
[3] Department of Child Studies and Social Work, Mount Royal University, Calgary, AB T3E 6K6, Canada
[*] Correspondence: pchoate@mtroyal.ca

**Abstract:** Fetal Alcohol Spectrum Disorder (FASD) is considered a lifelong disability that has been framed with neurobiological descriptions focused on the brain. These are important features but fail to tell the story of living with FASD. By surveying those with FASD, this work expanded upon prior survey work which illustrated a multitude of early-onset physiological issues occurring at rates much higher than is typical of the general population. The current project, again using an anonymous survey methodology, sought to open up other direct experiences to better understand the complexity of living with FASD. An anonymous online survey was used to gather data on adversity in childhood, schooling, employment, housing and finances, and involvement with the criminal justice system, as well as relationships and parenting. Results indicate high levels of adversity throughout the life span; vulnerability to manipulation, which is connected to involvement in the criminal justice system; struggles with housing; economic instability; and struggles maintaining employment, as well as difficulties with social and familial relationships. Systemic stigma was also identified. Suggestions are offered to inform others on how support can be enhanced and targeted with a goal of improving quality of life, as well as dealing with self-imposed stigma. The survey was developed by adults living with FASD who have served as a long-standing advocacy and educational group influencing policy and practice in the field.

**Keywords:** Fetal Alcohol Spectrum Disorder; FASD; FASD living experiences; anonymous survey; stigma and FASD; self-stigma and FASD

## 1. Introduction

Fetal Alcohol Spectrum Disorder (FASD) is a lifelong disorder that impacts both brain and body functions. Regrettably, it has often been framed as having levels of impairment that preclude pro-social functioning, although more recent research has shown this to be a significant overgeneralization, tending towards stigmatization [1,2]. In addition, more recent research shows that FASD exists across a broad range of expressions and thus also capacities. In addition, strengths may lie in various domains while limitations may exist in other domains, with experiences being quite uneven across the FASD population. There is not a universal or common expression of FASD; rather, it is heterogenous [3]. The rates of FASD in the general population in North America now appear to be 4–5%, inclusive of various expressions and intensity of FASD [4,5]. There is also an unknown number of people who are not diagnosed but who believe they are experiencing the effects of perinatal alcohol exposure [6]. This can occur for a variety of reasons, such as lack of confirmation of alcohol use in pregnancy, which partially arises out of failure to ask about alcohol use during pregnancy care, and lack of access to assessment services, as well as a misattribution of observed behaviors that are labeled in some fashion as a "badly behaving person" as opposed to inquiring into why the behaviors exist [7,8].

This paper presents a secondary data analysis of anonymous survey responses of people living with FASD in what might be euphemistically called the real world. The goal of the paper is to highlight how life is experienced by those living with FASD. There are very few data available that consider voices articulating such daily experiences, including interactions between the person with FASD and health care, education, criminal justice, and the community. This work builds on a previous survey that looked at the whole-body physiological experiences of FASD, which "... dramatically highlights the significant adverse effects of prenatal alcohol exposure on long-term vulnerability to disease and disorders over the life course, above and beyond what has traditionally been described in the literature." [9] (p. 211). Thus, FASD can be seen as a "whole body" disorder as opposed to one that is restricted to the brain and behavior.

The current project is a logical follow up to better understand the experiences and challenges of those living with FASD. It allows those living with FASD to give voice to their own experiences.

## 2. Literature Review

Fetal Alcohol Syndrome (FAS) seemed to enter the medical lexicon following the 1968 work of Lemoine et al. [10], although, as Brown et al. note, awareness of concerns with alcohol consumption and pregnancy date back beyond the 1700s [11]. Jones [12] provided one of the earliest detailed examinations of its morphological and developmental features. This was followed closely by a paper in which Jones et al. [13] further explored what they termed to be "the first reported association between maternal alcoholism and aberrant morphogenesis in the offspring" (p. 1267). Both articles remain prominent in the literature, each having been cited well over 3000 times. It may well be argued that Jones et al. [12,13] laid the foundation for how FAS would be seen for years. Even now, FASD may still be referred to as FAS, along with related terminology such as Alcohol-Related Neurodevelopmental Disorder (ARND), Alcohol-Related Birth Defects (ARBDs), and neurobehavioral disorder associated with prenatal alcohol exposure (ND-PAE). The term Fetal Alcohol Effects (FAEs) was also used to describe a child with intellectual disabilities connected to alcohol consumption in pregnancy [14]. Today, FASD is recognized in medical, nursing, midwifery, social work and psychology, criminal justice, and other sociologically related disciplines, although the depth of that awareness is still wanting. However, that should not lead to a conclusion that there is a broad understanding of what it means to live with FASD [15–17].

FASD has now come to be known as an "umbrella term" that considers impacts on social, behavioral, physical, and cognitive aspects of a person's life, although there are expressive variations across those impacted. Thus, there is no "one" presentation. It also exists across a spectrum. Those with similar areas of impact may express them in very different ways. Furthermore, the expression of FASD in an individual's life tends to shift over the life span [18]. In this respect, life course [19] theory assists us in understanding how living with FASD has many manifestations. Not only is there no one form of FASD, but there is also no one life course pathway.

The diagnosis itself brings stigma (defined as negative and unfair beliefs about people with FASD) which prejudges the person, narrowing the true experience of living with FASD. The prejudgment and a stereotypical view of incapacity versus capacity create a narrative that may have little to do with reality [20]. When that happens, an authentic understanding of what it means to live with FASD is lost [21]. This survey seeks to report on the truth of the life being lived, drawing upon actual experiences which can illustrate aspects of success but also the many challenges that occur throughout the lifespan [22]. These authors identify clusters of living experiences which include compounding stigma, environmental adversity (such as early life experiences, problems with social determinants of health such as poverty, lack of housing, lack of access to physical and mental health services, and adversity from a variety of events across the life course), and co-occurring

disorders including neurological development, mental health, and substance abuse, as well as challenges with family functioning.

McLachlan et al. describe that difficulties in daily living can be seen across the life span [23]. They describe these occurring in a number of domains, including independent living needs, substance abuse, employment instability, legal problems including victimization, trouble accessing stable housing, and disruptions in education. This can be further complicated by the lack of support for youths transitioning to adulthood [23–25]. Life course challenges occur not only in daily living but across time and in multiple life domains [2,23].

A better understanding of how life is experienced by those with FASD has the potential to increase understanding and subsequent support in both community and professional settings. In turn, this has the potential to increase the quality of life for a person living with FASD [24]. An example of this is seen in the results associated with Adverse Childhood Experiences (ACEs), which is an important way to consider the foundations of strength, resilience, and risk as people enter their adult years from age 18 onwards. The ACEs consider "the relationship of health risk behavior and disease in adulthood to the breadth of exposure to childhood emotional, physical, or sexual abuse, and household dysfunction" and the impact on later life functioning [25]. Kambeitz et al. [26,27] have shown that the linkage between FASD and elevated ACE scores increases the risks for comorbid neurodevelopmental disorders. Understanding the degree of exposure to ACEs in addition to the FASD is vital so that risk minimization and resilience building can occur.

When looked at from a life course perspective [28,29], people with FASD often face not only individual event traumas but also cumulative ones arising from ACEs, as well as factors that may not be measured in that way. This can include experiencing SDH that adversely affect life outcomes, early life interruptions such as involvement with child intervention services and placement in foster care, often with placement instability, difficulty with establishing social and peer relationships, and challenges with integration in school and staying on the same developmental pathway as other children. Individuals with FASD can be vulnerable to negative peer influences due to social isolation, as well as challenges with appropriately judging the nature of what is sought in social situations and difficulty assessing risks, increasing victimization [30]. The desire to belong is powerful. This can lead to poor social decision making, leading to involvement in the criminal justice system [24].

While negative experiences for people with FASD are common, the value of protective factors should not be lost. These include early diagnosis and support, stable caregivers, and educational, mental, and physical health systems that understand the nature of FASD and what is needed for a person to successfully face life's challenges. Transitioning into adulthood brings its own challenges, including having to interact with systems that are based upon chronological age rather than developmental capacity. This means that service systems expect adult capacity which a person living with FASD may not have developed. Adult-focused FASD services are even less available than childhood services, particularly away from larger centers [29,30]. The support plans that serve the individual best are based on what is possible and the recognition that time-limited programs are counterproductive. Good programming occurs over extended periods with persistent belief in the possible.

A major barrier in the lives of persons with FASD is that systems serve systems. Eligibility requirements to access services fail to consider the realities of living with FASD. For example, it may seem equitable to make all applicants for a service have a full-scale IQ of less than 75, despite evidence that those with FASD can have scattered profiles that can suggest higher IQs which do not reflect functional capacity. In order to be successful, equity requires that applicants for a service are considered based upon need as opposed to arbitrary classifications that fail to consider the truth of living. There is no equity, for example, when a person living in some parts of Canada has access to diagnostic and support services while those living elsewhere do not, even though theoretically everyone may be eligible for diagnostic services [31,32]. A further example is that assessment and

diagnostic services are more focused on children. Adult clinics are few and hard to access, which creates systemic discrimination for an adult. This can be seen in the justice system, as the lack of a diagnosis makes it very hard for a court to take FASD into consideration, which can then negatively impact sentences [33]. The current work seeks to increase knowledge about what it means to live with FASD.

## 3. Methodology

An anonymous survey was made available to people with FASD. This was done through community agencies and self help and support groups mainly across North America via an online link. There were two periods open for responses—August 2019 to February 2020 and March 2020 to December 2020. All respondents had the same survey available. In total, there were 468 responses analyzed for demographic and thematic purposes. Respondents were assured that there was no way to link responses to their identities. Not all respondents answered all questions, which creates some variation in response rates per question, meaning that the *n* for responses varies from question to question.

The survey was developed by the Adult Leadership Committee, which is a network of people living with FASD. The areas of inquiry were drawn from their living experiences.

This is a secondary data analysis of anonymous information. The Human Research Ethics Board at Mount Royal University determined that this did not require ethics clearance [34].

## 4. Results

### 4.1. Demographics

Survey respondents identified as 43% male and 55% female, with 2% as other or prefer not to say. The average age was 30 years. A total of 73% reported an IQ of 70 or over, while the remaining 27% reported below that level. Just under half of the respondents (49%) were from Canada, while 37% were from the USA, with the balance from other parts of the world. A total of 45% of the respondents indicated that they completed the survey on their own, while 55% received help from another person, mainly a parent. Table 1 reports the diagnostic categories.

**Table 1.** Self-reported understanding of diagnostic position.

| | |
|---|---|
| Fetal Alcohol Spectrum Disorder (FASD) | 28% |
| Fetal Alcohol Syndrome (FAS) | 24% |
| Alcohol-Related Neurodevelopmental Disorder (ARND) | 14% |
| Other * | 6% |
| Partial FAS (PFAS) | 5% |
| Fetal Alcohol Effect | 3% |
| I think I have FASD | 20% |

* Included static encephalopathy; FASD with sentinel features; Neurobehavioral Disorder PAE—this level of undiagnosed but suspected FASD is not uncommon, as the rate of FASD is believed to be significantly higher than the diagnostic population suggests. (1) May et al. [4] suggest that in the US, the rate is 31.1 to 98.5 per 1000 children. This is compared to older data indicating 10 per 1000 children. (2) Popova et al. [5] report Canadian rates ranging from 18.1 per 1000 up to 29.3 per 1000.

Of the 371 who reported a diagnosis, 32% were diagnosed at 5 years of age or under; 20% from ages 6 to 10 years; 18% from 11 to 15 years; 13% from 16 to 19 years, and the balance (16%) at age 20 and over. Such a wide range of chronological and developmental stages for assessment may impact expressions of both risk and protective factors.

### 4.2. Sense of Self

This section refers to survey questions which focus upon how respondents perceived being understood or misunderstood and included or excluded, as well as the degree to which they had a sense of belonging. On average, responses reflected an overall sense of not being understood and more often being excluded. The responses also showed an internal sense of inferiority. One of the more worrying features is that the majority reported not having the FASD diagnosis explained to them post-diagnosis (see Figure 1).

**Perceived Undertsanding and Inlusion in FASD**

**Figure 1.** Having connections to people with FASD who are acting as pro-social supports at a peer level is an important informal linkage that can be sustained over time.

In Table 2, respondents reported a desire for connection, particularly with those who are also diagnosed. Having relationships with others facing similar life experiences can also be a way to have affirming connections to validate self-experiences and self-worth.

**Table 2.** Respondents' desire for improved connections within an FASD community.

| **Desire for Connection Through...** | |
|---|---|
| Would like to attend a conference just for those who have FASD | 76.9% (347/451) |
| Would like to know more people with FASD | 79.8% (360/451) |
| Felt it would be helpful to have had someone with FASD to talk with them as needed post-diagnosis | 86.9% (319/367) |

### 4.3. Adversity

As noted above, adversity in childhood can have a significant impact on functioning across the lifespan [4,5]. The following are some of the key findings concerning adversity across the lifespan. The following results show that respondents experience high levels of life adversity. These results are consistent with those of Flanigan et al. [35] and Tan et al. [36].

Figures 2–6 illustrate the prevalence of adversity. The most commonly reported adverse experiences before 18 were verbal abuse (58%), lived with anyone in the home who drank alcohol heavily or was an alcoholic (54%), and lived in a home with someone who was depressed, was mentally ill, or attempted suicide (53%). After the age of 18, the most commonly reported adversities were being manipulated by others to do what they want (83%), intimidated or threatened (74%), and talked into doing something that was wrong (69%).

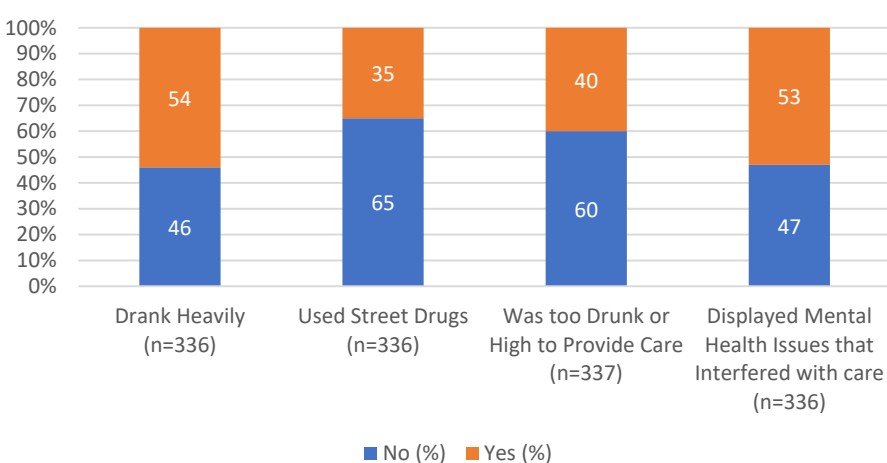

**Figure 2.** Examining the exposure to ACEs across various caregiving environments. Respondents may have answered for one, two, or three living situations. (Mental health included depression, suicide, exhaustion or other mental health issues).

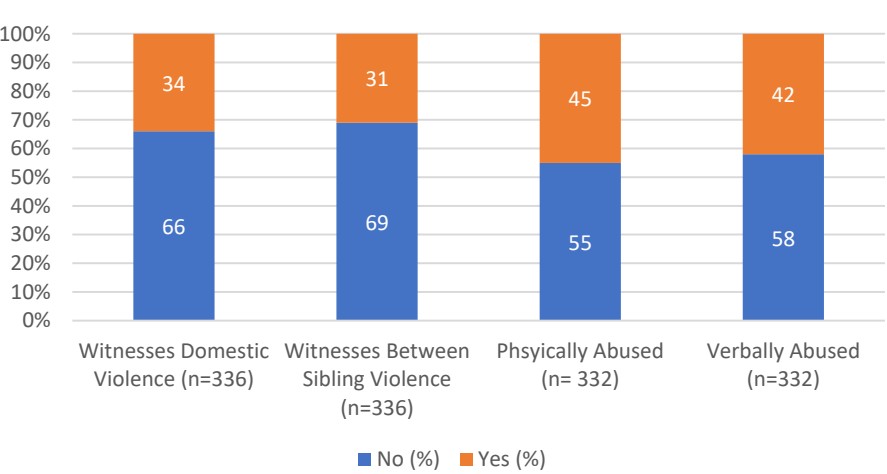

**Figure 3.** Exposure to interpersonal violence within the home.

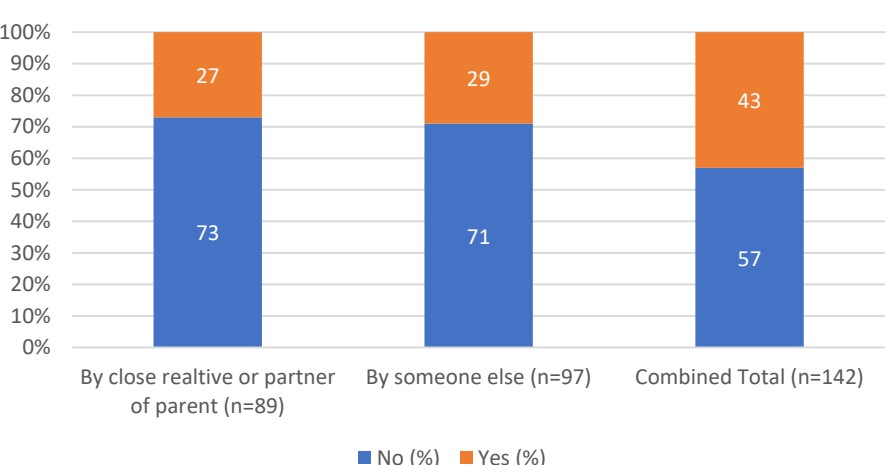

**Figure 4.** Sexual abuse reported by respondents. As indicated in the chart, 44 individuals reported sexual abuse in both categories.

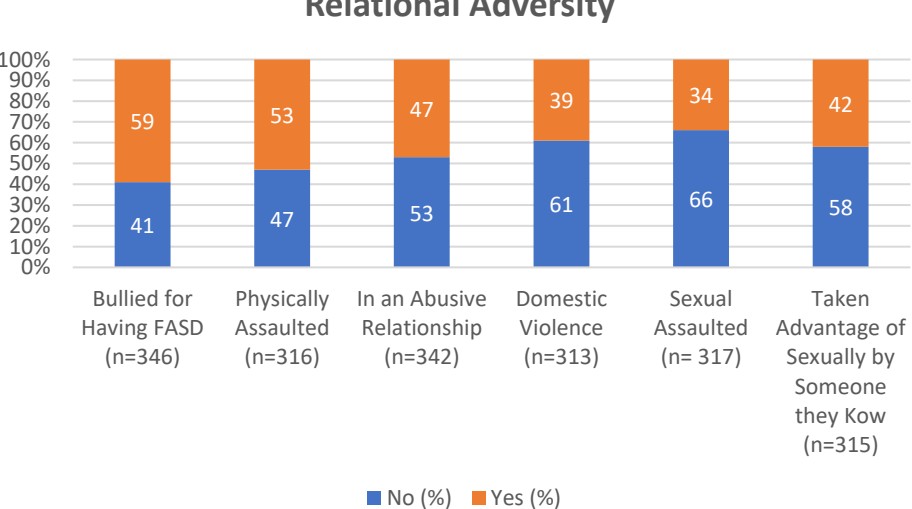

**Figure 5.** Adversity of physical abuse and sexual assault—interpersonal violence. Numbers may vary from Figure 4 due to varying response rates.

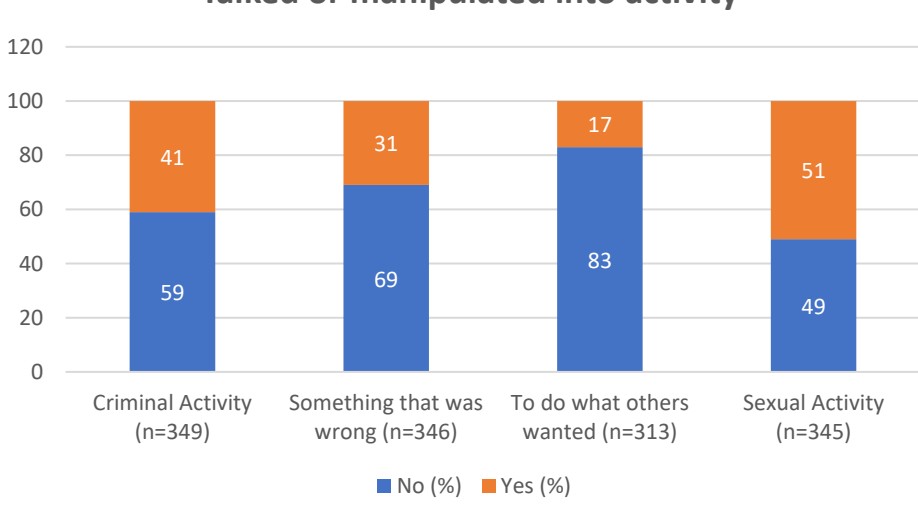

**Figure 6.** The behaviors reported linked to manipulation.

Figure 2 helps us to see the linkages between caregiving experiences of substance abuse and mental health within the lives of respondents. Trauma from one generation to the next is an outcome of such exposures, which has been documented throughout the population with FASD [2]. Future research is needed to parse out these experiences to better understand the depth, duration, and frequency, as well as the presence of any protective factors. Such information has the potential to improve interventions.

As seen in Figure 3, the majority of respondents have exposure to some form of interpersonal violence in the home environment, although this does not include coercive control [37]. This may be a rich area for exploration, particularly given the results noted later in Figures 4 and 5.

Figure 4 shows what might be best described as an alarming rate of sexual abuse victimization within the respondent population at levels not typical of the population at large [5,36]. Several considerations arise from these results, including how many of these cases have been disclosed. The vast majority of sexual abuse victims do not disclose [38]. In addition, given the data on manipulation in Figure 6 below, further research in this area is needed. Disclosure might be as high as 1:5 across the lifespan as opposed to proximal to the event. Victimization in childhood raises the probabilities of such in adulthood [36,37]. Combined with increased risk for manipulation, this is worrying.

The data in Figure 5 offer some more complex understanding of the rates of IPV. These results, along with others noted above, reinforce that persons with FASD are at much higher risk for victimization.

### 4.4. Education

Also related to quality of life is the participants' education history. This survey gathered information on educational completion, educational support, and employment history related to post-secondary education. Of those who answered that they attended high school or were currently in high school (*n* = 423), 26% attended or had attended a regular program without support in high school at the time of the survey, 32% were enrolled or had been enrolled in a regular program with an individual educational plan (IEP) at the time of the survey, and 40% were or had been enrolled in a special education program or school. Of those not currently in high school (*n* = 351), 29 percent did not finish high school (*n* = 100), and 77% of those who did not finish (*n* = 77) said they believed they could have if there had been more help. Eighty-three percent of those surveyed who were no longer in high school said that, looking back, they did not understand what was taught to them in class. Concerning post-secondary education, which was denoted by job skill programs, career training programs (i.e., technical and trade schools), and a college or university degree program, the survey asked if participants had received a job, followed by their current employment status. Of the 163 who said they attended a job skill program, 74% received a job and 31% still had a job. Of the 101 who attended a career training program, 67% completed the program, 45% received a job, and 26% still had a job. Finally, of the 155 who attended a college or university degree program, 48% graduated, 37% received a job, and 18% still had a job in their area of study. These data highlight the need to support the education process over the long term to enhance outcomes.

### 4.5. Employment

At the time of the survey, 49% (291/342) indicated being employed full or part time. They indicated that they found working a struggle, identifying such factors as being overwhelmed, being worried about doing the job properly, being too tired to do other things, and deteriorating physical and/or mental health. The social relationships that form part of a work environment were often seen as stressful. Indeed, the overall stress of being employed meant that 63% (200/317) of respondents reported they could only work part time. Just over 80% (265/319) indicated they enjoyed working. Seventy-one percent (196/276) indicated that they felt good, valuable, and productive, like others in society, when they were working.

Consistent with the perception that being a person with FASD would often bring stigma, shame, and/or fear, 61.9% (195/315) reported they would keep the diagnosis secret from an employer, and 54.2% (137/253) reported not letting anyone at work know. Seventy-seven percent (248/322) believed they would not or probably would not have been hired if the employer knew of the FASD.

Being fired or laid off was a common experience, with 66.8% (194/290) describing this experience and 45.8% (133/290) stating it had occurred three or more times. Quitting was also common, with 76.9% (220/286) having done so. Just under 40% (114/286) reported having done so three or more times. The common issues related to sustaining employment included being overwhelmed; things going wrong in other parts of their life; worrying about performing the job properly; being too tired to do other things; physical and/or mental health growing worse; and struggling to get along with co-workers. This also illustrates the need to engage employers' understanding of the nature of FASD and how to support the needs of employees with FASD.

Of the 142 employed at the time the survey was completed, 199 (71.3%) reported earning less than CAD 1500 per month.

### 4.6. Finances

Just over half of respondents (51.9%) (240/462) indicated they received some sort of financial assistance from a government. Of those, 90% (216/240) indicated they received less than CAD 1500 per month. Of those receiving such money, three quarters reported this was not enough to live on.

Not surprisingly, financial limitations also impacted having enough money to cover expenses over the course of a month and access to healthy food (as defined by the respondent), as well as difficulty affording medications. Notably, 77.8% (302/388) indicated that they regularly receive financial help from family or others to help address needs such as rent, groceries, and phone bills. These financial limitations add to levels of dependency, with only 1:5 being able to live independently. This highlights that social support systems are likely inadequate most of the time.

### 4.7. Housing Instability

The survey asked the respondents about their living situations (i.e., where they lived, how housing was paid, who they lived with, etc.). At the time of the survey, 5% (21/419) were homeless and 25.9% (114/439) had experienced eviction at some point. This was connected to challenges, with 38% (155/407) reporting problems paying rent or not having enough money to pay the rent and 51.1% (208/407) struggling to remember to pay their rent on their own each month.

### 4.8. Memory Issues

Respondents indicated a number of concerns arising from memory problems. Without help, common concerns included remembering to take medications, pay bills and rent, and take care of personal and household hygiene, as well as eating.

Respondents described being unable to remember to do things without help, such as paying the rent (50.9% (188/369)) or bills (73% (233/319)), taking medications (59.9% (170/284)) and refilling prescriptions (59% (163/276)), doing laundry (58.8% (193/328)), or cleaning house (73.8% (240/324)). This need for help also included personal hygiene: showering (30% (98/326)), cleaning teeth (54.7% (178/325)), washing hair (36% (117/325)), or grooming hair daily (37% (114/308)).

### 4.9. Family, Partners, and Parenting

Only 17.2% (61/323) of respondents were raised by their birth family. Otherwise, they were raised in quite a variety of living situations. A total of 26% percent (85/323) reported one long-term foster care family, while a similar number of 26% (84/323) reported being raised in a number of family and foster care arrangements, creating significant instability. A further 31.5% (102/323) described adoption. A significant portion of respondents, 82.7% (292/353), had birth siblings, although it is unclear what the long-term relationships with them might look like.

Of those responding in this area, 42.9% (147/342) were in a partner relationship at the time of the survey. Fourteen percent were married. We are missing a sense of the relational stability, which would be worthy of further investigation.

In terms of being a parent, 28.9% (100/346) had children, roughly split half and half with the child living or not living with the parent with FASD. The other living arrangements were with the other parent or family member, in foster care, or having been adopted.

### 4.10. Friendships

In terms of friendships, 65.7% (227/345) said that it is hard to make friends, while 81.7% (282/345) indicated that it is hard to keep friends. Being alone was preferred by 60.7% (209/344). Being taken advantage of people they considered friends was described by 80.8% (278/344), which may be related to the 75% (258/344) who indicated that they decide who is a friend too quickly. For the respondents, 77.1% (264/342) described that being with people is exhausting, and 71.8% (245/341) described that being with people also

brings anxiety and nervousness. A total of 66% reported that making friends is difficult, and 61% reported being happier alone. Half of the respondents had most of their social interactions online.

*4.11. Criminal Justice*

Involvement with criminal justice was fairly common. Of the 344 participants responding in this area, 38.9% had been arrested (98 charged and 59 convicted, some more than once). Just over 7% (25/344) reported they had been in a youth prison, and 11.3% (39/344) had been in an adult prison or jail. Respondents indicated a series of other challenges within the criminal justice system, which included agreeing with the police even though they did not commit the crime; pleading guilty without understanding the consequences; and being talked into committing crimes or forced to do so, which may be related to vulnerability arising out of the FASD.

Being a victim of a crime was common and was reported by 53.8% (184/342). The most common forms of victimization were from physical assault, sexual assault domestic violence, financial exploitation, and being robbed or mugged. As an indicator of the vulnerability of a person with FASD, 135/183 answering stated they did not report the crime, which may be related to the 76% (142/185) stating they have not been believed by the police in at least one encounter. Part of the challenge is being seen as a credible informant. Understanding the process is a further challenge, with 65.2% (163/250) stating that when they did have encounters with the police, they did not understand the process [39].

All of the results related to housing, employment finances, social relationships, and involvement with the criminal justice system need to be considered with the earlier noted results related to adversity in childhood and adult years. These are intersectional.

## 5. What Could Help

The respondents identified a variety of ways they can be supported. These include mental health clinicians and other health care practitioners who specialize in FASD. Those might be thought of as more formal supports [40], but they also identified informal supports such as trusted people to speak or act for them—in other words, people who are informed and trustworthy and who listen.

For the person living with FASD, both formal and informal supports need to show they accept this person as one with their own hopes and dreams. Based upon the suggestions from 313 respondents, below are our insights into how help might look and be structured, providing the basis for needed policy discussion.

1. Access to a mental health clinician who specializes in FASD.
2. Availability of a doctor or nurse practitioner who knows about FASD.
3. A person who can help when something goes wrong.
4. A person who can be trusted to give advice when needed.
5. Enough money to meet monthly needs.
6. Help with tasks of daily living such as cleaning and laundry.
7. Having a trusted person who, with permission, can speak and act for the person with FASD.
8. A trusted person to manage or help with money so that the person with FASD is less likely to be taken advantage of. This may also include being able to attend appointments so that there is someone present to support the person's understanding of what has been said and recommended.
9. Help to obtain and sustain employment (this would be a person who understands what is and is not possible).
10. The ability to engage in activities that are important to the person.

Arising from these data, an area that might be considered is the ways in which a person living with FASD might frame themselves. This is an area for further research. However, as the chart below shows, challenges with daily living can become incorporated

into a dialogue of self-stigma, as seen in Figure 7 below, which was developed with the Adult Leadership Committee from reflecting upon the results.

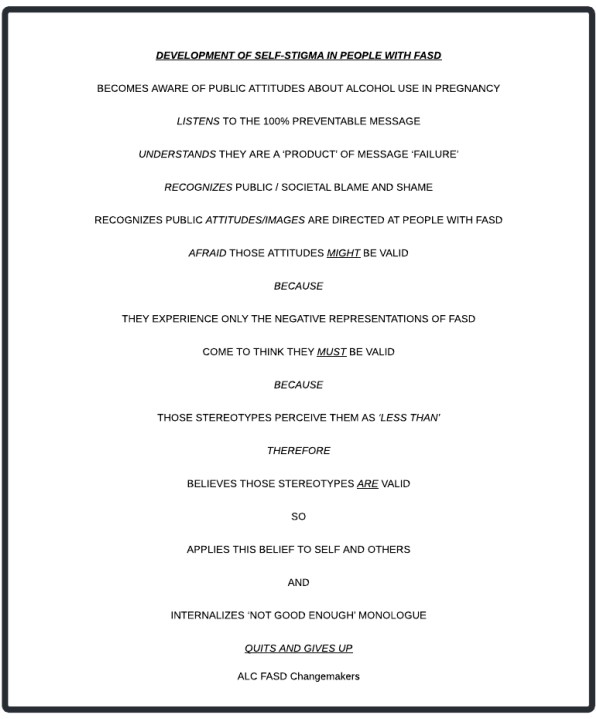

**Figure 7.** The evolution of the dialogue of self-stigma. (Source: ALC FASD Changemaker authors).

## 6. Discussion

Being diagnosed with FASD is a step in gaining recognition and understanding for one's living experience. As can be seen with these data, the diagnosis is an explanation, but it is not a predictor of the life course in and of itself. The data show ongoing, significant challenges for which an individual is likely to need ongoing support and connections across the lifespan. This includes having professionals who understand what FASD is about [7].

This research study was designed to explore quality of life among adults with an FASD. The results of this study are meant to lay the groundwork for richer and deeper research which might also focus upon more of the successes experienced, which the members of the Adult Leadership Team frequently report in their conference presentations. Similarly, it sought to explore the concepts of adverse life experiences from childhood forward, capturing an initial understanding of continuing stress-correlated experiences that may exist in adulthood and their reported rates. Previous limited research has explored ACEs in this same population, but to the authors' knowledge, adult experiences linked to childhood adversity have not been explored. What was notable was the rates of abuse, victimization, and homelessness not just before but also after the age of 18. The adversity results illustrate that adverse experiences indeed appear to continue well into adulthood. Given that the average age of the respondent was 30 years old, this pathway of ongoing adversity negatively impacts the quality of life for adults living with FASD. Such patterns of ongoing adversity leading to intergenerational trauma (IGT) should be considered [41]. This does not mean that all parents who have children with FASD are from a traumatized legacy. What we see in these data, however, is that many adults in the sample show IGT linkages. This is evident in the data.

This might be considered in terms of the exclusion in society also seen in these data, such as believing that they are misunderstood and being seen as intellectually and socially inferior. Social messaging within society and systems such as schools and jobs, combined with memory difficulties, result in feeling "less than". After the age of 18, several continuing experiences beyond abuse were explored and reported on the survey, such as being taken

advantage of, intimidated, or talked into an activity, revealing these to be areas of concern for an adult with FASD.

Yet, to use an incapacity lens limits the notion of what a person living with FASD might accomplish. As adults, many living with FASD pursue parenthood but are often quickly perceived as incapable, thus coming to the attention of child protection [42]. Once a person with FASD enters the child protection court system, they are almost always likely to lose custody of their children [43]. Nearly 74% report having children, and in spite of work-related stigma, many pursue employment, with over half (62%) reporting they keep their diagnosis a secret from their employer. These are also areas for further research so that we better understand the complexity and depth of the life course experience.

As has been discussed above, being vulnerable to manipulation creates a number of risks of victimization, as well as becoming involved in the criminal justice system, where a person with FASD may struggle to explain their role effectively. When looked at as a cohesive set of data, the material illustrates the degree of exposure to trauma across the lifespan. There is a substantial body of work that shows trauma impacts a person in multiple ways (physically, emotionally, and socially, for example), in ways that are both cumulative and persistent [40,41]. These are life experiences to which a person with FASD appears to be more vulnerable. This emphasizes the need for intervention, support, and healthy connection on an ongoing basis.

This study lays the groundwork for richer exploration or understanding of what appear to be disparities and potentially higher levels of stress and lower levels of quality of life within the adult FASD population. ACEs such as trauma, abuse, neglect, etc., have been shown throughout the literature to negatively impact development across the lifespan [42], but what happens as continuing adverse experiences happen in adulthood? What this survey demonstrated was high reporting levels of the perception of stigma and lower levels of support. The key areas covered in this paper, from housing, education, and employment to criminal justice, all indicate subareas where services could be developed. This includes scaffolding of services over the life span. It is equally valid to prioritize dismantling systematic misunderstanding of what might be possible, opposing stigmas which serve as powerful barriers.

People living with FASD do not seek to hide their realities; rather, they seek to have them known, even though they feel the need to hide their diagnosis in order to be given opportunities [9,20]. When they are allowed to be open or feel safe to do so, the genuineness of relationships with caregivers, professionals, and the community at large remains very possible. They should also have the right to tell their story, as it is *their* story to tell [44–46]. Stereotypes lead to stigmatization arising from a single story of incapacity which flattens the experiences, capacities, and opportunities of persons with FASD [8]. An alternative narrative is that people with FASD have the right to thrive [22]. They also have the right to be heard, according to Reid et al.; many in the FASD community state "Nothing about us without us" [18]. This informs the notion that people living with FASD offer valuable insights which serve to educate formal and informal support systems working in partnership with them.

As Aspler et al. [1] note, stereotypes are powerful and persistent, creating a narrowing view of the person. A balanced view of life matters so that support and interventions can be crafted to the reality of strengths, weaknesses, and limitations and opportunities. The public perception of FASD is heavily driven by the stereotypical presentations often seen in various forms of media [8]. Regrettably, the narrow, stigmatized story of living with FASD becomes "*the*" story, which interferes with the ability of the living truth to be told and of people to have supportive places in society. People with FASD are also less accepted than those other disabilities, such as those with Autism Spectrum Disorder, who also experience a range of behavioral, cognitive, emotional, and physiological experiences [8,47]. Family life is also impacted by FASD and ASD, although ASD is seen as more receptive to intervention [1,47,48].

In Vancouver, Canada, a series of international conferences regarding FASD were held from 1987 to 2019 which included an Adult Leadership Committee composed of individuals with FASD. COVID-19 disrupted holding the conferences, which offered professional and academic presentations as well as opportunities for those with FASD to provide leadership, share living experiences, support research, and connect. As can be seen from the results in Table 2 above, these types of opportunities are sought after and have a high probability of being pathways to better life course decision making. The conferences are returning in 2024 and will be held in Washington state.

## 7. Limitations

By definition, anonymous survey data have limitations. The respondents are self-selecting and may not be representative of the population with FASD. There is no verification of self-reported information, and some respondents who believe they may have FASD could, if properly assessed, turn out to not have the diagnosis. In addition, there is no capacity to clarify, follow up with, or expand upon responses. The survey may not be representative of the population with FASD, given that the survey required access to online platforms and technology. On the other hand, respondents can be quite sure that their information will remain anonymous. This protects them from accidental disclosure, as well as limiting the fear of embarrassment, shame, or guilt that may arise from describing life experiences.

## 8. Areas for Future Research

This work opens up ways to think about living with FASD. It invites substantial follow up to dive deeper into intersectional, complex experiences so that the texture and variations of living with FASD can be better understood. Qualitative work, which might include phenomenology, could add rich storytelling. Grounded theory approaches could thematically explore these stories, while further quantitative work could add more detail to the areas explored here. While there is an IGT linkage found in the data, the data do not distinguish between a positive correlation between higher rates of IGT and higher rates of ACEs-A. Further research would need to be conducted to answer such a question. Similarly, more research needs to be conducted to explore whether a lack of resources in adulthood exists, and if so, whether there is a relationship to ACEs-A.

Future research might consider exploring coercive control as a factor given the growing body of work around this aspect of interpersonal violence (IPV), particularly Barlow and Walklate's [37] work showing that those with FASD are more vulnerable to manipulation, a core feature of coercive control.

Given the significant power of relationships, we need to better understand how to support adults with FASD when their parents/caregivers pass away. What support will then be available, and how will individuals with FASD maneuver their way through complex systems with variable eligibility requirements?

Taking the time to understand what works is often missing from research agendas, as the research is often deficit-based. Understanding strength-based perspectives would allow for a richer understanding of how people with FASD can intersect with systems without being prejudged.

Other research avenues might consider how to implement the ideas reraised in this work, validating, for example, systemic life course interventions, impacts arising from education, and other approaches to reducing stigma in the public and professional arenas. Other research might begin focusing upon altering the pathways to self-stigma.

One particular focus of future research is to continue to find ways to include the voices of those living with FASD. In particular, a more focused look at capacities and strengths might be developed. Even so, these data allow for a look into the challenges faced by those living with FASD and what must be managed on a daily basis. The challenges are not the whole story [49], and the literature in that sphere is quite limited.

## 9. Conclusions

These anonymous survey data have opened up further understandings about living with FASD. In particular, they show the degree to which Adverse Childhood Experiences and Adverse Adult Experiences are common. Highlighted as well is the vulnerability that can be seen in being manipulated and how that contributes further trauma to living with FASD.

**Author Contributions:** P.C. and E.H. wrote the preliminary and final submission versions of the paper. C.J.L., K.G., M.H., J.M. and A.L. conceived the original survey and arranged for making it available and gathering the data. They also performed the original data analysis. P.C. and E.H. completed the secondary data analysis. All authors were engaged in reviewing, editing, and recommending changes to the paper. All authors have read and agreed to the published version of the manuscript.

**Funding:** This research received no external funding.

**Institutional Review Board Statement:** Not required as per TCPS-2 standards for anonymous surveys. Secondary data analysis was approved by the Human Research Ethics Board of Mount Royal University.

**Informed Consent Statement:** Not applicable.

**Data Availability Statement:** Data are contained within the article.

**Conflicts of Interest:** The authors declare no conflicts of interest.

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
