# Peer review of "FASD: The Living Experience of People with Fetal Alcohol Spectrum Disorder—Results of an Anonymous Survey"

_disabilities, doi:10.3390/disabilities4020022_

Round 1
Reviewer 1 Report
Comments and Suggestions for Authors
The present paper, “FASD: The Living Experience of People with Fetal Alcohol Spectrum Disorder—Results of an Anonymous Survey”, presents a secondary analysis of anonymous survey data. It provides a rare and much needed perspective regarding FASD and important insights into the life experiences of persons living with FASD. I look forward to seeing this published!
General comments
1. Organization of manuscript: This manuscript takes a somewhat non-traditional approach to manuscript content organization. In some places the authors may want to hew closer to the usual format for the sake of clarity: methods describe what was done and how, the instruments used (your survey), and how you performed statistical analyses, handled missing data, etc. All the results are presented in the results. Interpretations of the results are discussed in the discussion.
2. The first paragraph of the literature review uses the term FASD but concerns only FAS (Fetal Alcohol Syndrome), the portion of FASD diagnoses where the characteristic facial dysmorphology is present. Lemoine was specifically investigating 127 children of alcoholic parents and Jones and Smith coined the phrase Fetal Alcohol Syndrome and identified the association of chronic alcoholism to a distinct pattern of effects. The term FASD, which indicates a “spectrum” of conditions, did not enter the medical lexicon until much later: after we recognized that prenatal alcohol exposure, not alcoholism, was the critical focus. FASD is not yet a broadly recognized medical diagnosis, unlike FAS, although we hope it soon will be. (I agree that the term is unfortunate but it’s harder to blame Jones and Smith when we know how it came about.)
3. Methodology – This section currently presents results (468 responses, male/female, table 1, etc.). Consider moving all results to results section. In methods, please describe recruitment (how promoted, how people accessed the link, inclusion/exclusion criteria), the survey, possibly with examples, statistical analysis (how each number was calculated – this will help clear up why numbers differ in some figures).
4. Results – It would be helpful to start with total n recruited and the demographics in the results section. Table 1 could be demographics which would help the reader understand the results.
o Consider using the format “(n=numerator/denominator)” instead of just one number for clarity throughout. For example, 36% of 468 is not 396 (finances).
o There is discussion (sometimes with references) throughout the results section. This is non-traditional and some readers may miss information if not in the expected section.
5. There is overlap among table 3 and several figures following table 3. The table is more confusing than the figures because it may be misinterpreted to indicate that everyone among those who responded endorsed the question and leaves the reader to wonder whether there was an option to not endorse. Consider including those answering no in the table. Alternatively, consider presenting the findings once to avoid redundancy and for clarity.
6. The important section line 396 - 400 deserves its own heading and to be moved above “What would help?”. If you want to keep it where it is, consider rephrasing it to reflect how the pathways can be interrupted. Also, consider presenting this pathway in a more visual way – flowchart or illustration with arrows and interrupters.
7. Many “also”, “as well”, “even so”, etc. can be removed, but are fine if heavy use is your stylistic preference
Specific minor comments are indicated using comments in the attached manuscript.

Author Response
Reviewer 1
Thank you so much for your comments on the manuscript. These are very useful and supportive of the work. We truly appreciate the time you have taken.
- Organization of manuscript: This manuscript takes a somewhat non-traditional approach to manuscript content organization. In some places the authors may want to hew closer to the usual format for the sake of clarity: methods describe what was done and how, the instruments used (your survey), and how you performed statistical analyses, handled missing data, etc. All the results are presented in the results. Interpretations of the results are discussed in the discussion. We have taken steps to move the suggested information in places that made sense to us bearing in mind your comments. We hope you find that we have struck the right balance.
- The first paragraph of the literature review uses the term FASD but concerns only FAS (Fetal Alcohol Syndrome), the portion of FASD diagnoses where the characteristic facial dysmorphology is present. Lemoine was specifically investigating 127 children of alcoholic parents and Jones and Smith coined the phrase Fetal Alcohol Syndrome and identified the association of chronic alcoholism to a distinct pattern of effects. The term FASD, which indicates a “spectrum” of conditions, did not enter the medical lexicon until much later: after we recognized that prenatal alcohol exposure, not alcoholism, was the critical focus. FASD is not yet a broadly recognized medical diagnosis, unlike FAS, although we hope it soon will be. (I agree that the term is unfortunate but it’s harder to blame Jones and Smith when we know how it came about.) Your comments about Lemoine, Jones and Smith are well offered. We have changed the language in line with this comment. We have also added a bit more about the evolving language.
- Methodology – This section currently presents results (468 responses, male/female, table 1, etc.). Consider moving all results to results section. In methods, please describe recruitment (how promoted, how people accessed the link, inclusion/exclusion criteria), the survey, possibly with examples, statistical analysis (how each number was calculated – this will help clear up why numbers differ in some figures). We have moved the results from methodology to results. Thank you for the suggestion.
- Results – It would be helpful to start with total n recruited and the demographics in the results section. Table 1 could be demographics which would help the reader understand the results. There is not an n for recruitment as, being an anonymous survey, the invitation was broadly made throughout the FASD community. This is noted in the methodology.
o Consider using the format “(n=numerator/denominator)” instead of just one number for clarity throughout. For example, 36% of 468 is not 396 (finances). We have changed the format throughout to reflect that suggestion. Thank you, it does make the data clearer.
o There is discussion (sometimes with references) throughout the results section. This is non-traditional and some readers may miss information if not in the expected section. We have sought to modify that although some was kept for contextual reasons.
- There is overlap among table 3 and several figures following table 3. The table is more confusing than the figures because it may be misinterpreted to indicate that everyone among those who responded endorsed the question and leaves the reader to wonder whether there was an option to not endorse. Consider including those answering no in the table. Alternatively, consider presenting the findings once to avoid redundancy and for clarity. For clarity, we have removed table 3.
- The important section line 396 - 400 deserves its own heading and to be moved above “What would help?”. If you want to keep it where it is, consider rephrasing it to reflect how the pathways can be interrupted. Also, consider presenting this pathway in a more visual way – flowchart or illustration with arrows and interrupters.
- Many “also”, “as well”, “even so”, etc. can be removed, but are fine if heavy use is your stylistic preference
Specific minor comments are indicated using comments in the attached manuscript. We have gone through those and adjusted the manuscript accordingly
Reviewer 2 Report
Comments and Suggestions for Authors
Very important research paper to bring the voices of people with FASD to the literature. Thank you for this work.
Comments:
There are many typos, grammatical errors, spelling mistakes etc. throughout the paper. Please review and correct errors. I have identified some, but did not make notes of all of the errors.
Please check all references in text and in the reference list for accuracy.
Ensure that you include background literature on all areas that are discussed in the paper - employment, health care, criminal justice involvement, education etc. Need to lay the background literature for all the results.
Methods: add into the beginning of the methods section a statement about self-identifying FASD and the authors did not verify FASD diagnosis. Also, describe who created the survey and how it was developed.
Was there ethics for the origional survey? It seems that this survey was developed for the purpose of research and data collection and therefore ethics would be needed.
Discussion: the authors need to connect the findings with exisiting reserach and compare and contrast to existing literaure.
The beginning of the paper discusses strengths, yet strengths are not mentioned in the paper. The paper includes many different areas of focus, and it may be more helpful to narrow down the focus of the paper in order to create more purpose and direction.
Line 17: adversity typo
19: I think you need to identify the aim of the paper in a more clear way. Quality of life is very broad and general. What is the purpose of this paper and research?
35: run on sentence. Rephrase
55: add space before 1968
83: I am not sure what you mean by "accessing social determinants of health and adversity". Describe this concept further and how it relates to this paper.
153: "this secondary" - I am not sure what this means. Secondary analysis of data?
Figure 1: the description does not seem to match with what the figure is showing.
192: what evidence is there of a high probability of a path to life and decision making? please explain
218: add (IPV)
254: poor sentence structure. revise
292: this is repeated from above.
4.6. Memory was already described aboe in finances. Please revise.
431: typo FASD
Comments on the Quality of English LanguageMany grammatical, spelling, and writing errors throughout the paper.
Author Response
Very important research paper to bring the voices of people with FASD to the literature. Thank you for this work. Thank you so much for your support. As well, we want to thank you for the time you have taken to review and comment on this work.
Comments:
There are many typos, grammatical errors, spelling mistakes etc. throughout the paper. Please review and correct errors. I have identified some, but did not make notes of all of the errors. That is embarrassing and we hope our attention to this has corrected that.
Please check all references in text and in the reference list for accuracy. Thank you – we have done that.
Ensure that you include background literature on all areas that are discussed in the paper - employment, health care, criminal justice involvement, education etc. Need to lay the background literature for all the results. We have gone through the literature review and added a couple of new citations to ensure this is fully covered. Thank you.
Methods: add into the beginning of the methods section a statement about self-identifying FASD and the authors did not verify FASD diagnosis. Also, describe who created the survey and how it was developed. Thank you – we have added the verification issue in limitations. In methodology we have added a statement on the development of the survey.
Was there ethics for the origional survey? It seems that this survey was developed for the purpose of research and data collection and therefore ethics would be needed. The survey was initially done for the purposes of the adult leadership committee’s work within the FASD community, As such, in accordance with TCPS-2 Canadian ethics guidelines, clearance was not required. For this purpose, we clarified that with the ethics board at Mount Royal University, as a secondary data analysis, we were able to proceed.
Discussion: the authors need to connect the findings with exisiting reserach and compare and contrast to existing literaure. Thank you – we have added several aspects of this throughout the paper but particularly in the discussion.
The beginning of the paper discusses strengths, yet strengths are not mentioned in the paper. The paper includes many different areas of focus, and it may be more helpful to narrow down the focus of the paper in order to create more purpose and direction. We hope this has been successfully added in the introduction.
: I think you need to identify the aim of the paper in a more clear way. Quality of life is very broad and general. What is the purpose of this paper and research? Thank you – we have made this clearer
Reviewer 3 Report
Comments and Suggestions for Authors
The paper is presenting an anonymous online survey on adverse experiences of those with FASD in their childhood, schooling, employment, housing and finances, involvement with the criminal justice system as well as relationships and parenting. In total, 468 responses were analyzed.
I think, it is highly relevant to present the experience of those affected with FASD of what it means to live with FASD. Yes, there is no one form of FASD, and there is also no one life course pathway.
Regarding these facts, I wonder why authors present a study on adverse experience only. Yes, displaying the “Pathways for the Development of Self Stigma in People with FASD” is of major importance. Then again, the perspective focusing on deficits may be incomplete.
Patients and their parents or partners visiting our FASD clinic quite often refer to the positive aspects of FASD. Those with FASD are, for example, charming, making friends easily (ok, there is a risk in being open hearted and being easily talked into things). They are talented (artistic, doing sports…), they are skilled in manual work…
May be the positive aspects of FASD are ignored too often in studies on QoL in FASD. ACEs and self stigma definitely are burdensome, but may be they are not the whole thing.
Minor points:
Check the text (correct “data on advresity” etc.)
Check acronyms (e.g. ACEs-A vs. ACES-A)
Abstract: please add some results (insert them after methods) as “Results inform how supports can be enhanced and targeted with a goal of improving the quality of life” is not presenting results but conclusions.
In summary, this is a wonderful paper presenting the (or at least a) perspective of those with FASD.
Author Response
We are extremely grateful for your comments and the time you have spent reviewing.
The paper is presenting an anonymous online survey on adverse experiences of those with FASD in their childhood, schooling, employment, housing and finances, involvement with the criminal justice system as well as relationships and parenting. In total, 468 responses were analyzed.
I think, it is highly relevant to present the experience of those affected with FASD of what it means to live with FASD. Yes, there is no one form of FASD, and there is also no one life course pathway.
Regarding these facts, I wonder why authors present a study on adverse experience only. Yes, displaying the “Pathways for the Development of Self Stigma in People with FASD” is of major importance. Then again, the perspective focusing on deficits may be incomplete. This is a good question – at this stage the questionnaires were focused in these areas. A strength perspective does need to be developed and a further project may help along the way. We have added a note in the areas for further work.
Patients and their parents or partners visiting our FASD clinic quite often refer to the positive aspects of FASD. Those with FASD are, for example, charming, making friends easily (ok, there is a risk in being open hearted and being easily talked into things). They are talented (artistic, doing sports…), they are skilled in manual work…We agree and again this is an area that we should begin to more clearly explore. This is noted in areas for future work
May be the positive aspects of FASD are ignored too often in studies on QoL in FASD. ACEs and self stigma definitely are burdensome, but may be they are not the whole thing. We very much agree
Round 2
Reviewer 2 Report
Comments and Suggestions for Authors
Thank you for taking the time to address the edits and comments. A few more minor edits.
For the edits that were made, there continues to be some grammatical and punctuation errors. Please review the track changes made.
96: edit sentence "early life experiences, [povery, availability, lack] of social determinants of health... also make sure you define SDH if using abreviation later in paper
Add a transition paragraph at the end of the literature review to remind the reader of the aim of the study
Table 2 and throughout results section, use percentages instead of fractions. Fractions should be used in brackets
Adversity should be it's own section - i.e. 4.3
The title "FASD specifics" does not capture that section. Consider re-naming. E.g. sense of self
Put family, parenting, friends, and partners together in one section called family
Figure 1 - was this part of the study? If so, explain how it connects to the research conducted. If not, move to discussion
406: run on sentence.
Author Response
Thank you so much for the suggestions. We have incorporated them all with one minor exclusion - we elected to keep friendship separate from family, partners and parenting.
In terms of fractions / percentages - we agree your suggestion is a much better presentation.
Your support in reviewing the paper and offeriing suggestions has been valuable.